# OpenReview forum: "Do LLM Evaluators Prefer Themselves for a Reason?"
_ICLR.cc/2026/Conference — Submitted to ICLR 2026_

### Official Review · Reviewer_PZ79 · 2025-10-25

**Soundness:** 2
**Presentation:** 3
**Contribution:** 2
**Rating:** 2
**Confidence:** 3

**Summary:**

This paper addresses a timely topic: the self-preference of LLMs when acting as judges. Unlike prior studies, the authors conduct experiments on verifiable tasks (mathematical reasoning, factual question answering, and code generation), which allows them to distinguish whether an LLM’s self-preference is legitimate or harmful. The authors perform cross-family and cross-scale evaluations, where each model serves as both evaluator and evaluatee. They define three metric, SPR, LSPR , and HSPP, to quantify the magnitude and type (legitimate vs harmful) of self-preference. Empirical results show that stronger models exhibit higher self-preference, which is mostly legitimate. While harmful bias does exist, longer reasoning reduces them. Based on these findings, the authors offer several practical suggestions for improving the reliability of LLM-as-judge systems.

**Strengths:**

1. This paper studies a timely and relevant topic, the self-preference of LLM-as-judge, and provides practical suggestions for improving their reliability.
2. By focusing on verifiable tasks (math reasoning, factual QA, and code generation), the paper fills a gap left by previous works by allowing for subjective evaluation and comparison that distinguishes legitimate from harmful self-preference.
3. The conducts a systematic and large-scale evaluation across multiple model families and sizes. The comparisons are comprehensive, the experiments are well-organized and reproducible, and the presentation of results is clear and visually effective.

**Weaknesses:**

1. Some claims in the paper are subjective or speculative, not supported by empirical evidence. For example, in Line 252-256, the authors interpret correlations between model accuracy and self-preference as causal evidence that stronger models “recognize correct answers” or “identify bugs,” yet no qualitative analyses or case studies are provided to support these assertions. At minimum, the paper should include some examples showing how the LLM-as-judge identifies or explains specific errors in outputs to substantiate this claim.

2. This results have limited generalizability. While verifiable tasks allow for clear evaluation of LLM-as-judge behavior, most real-world applications of LLM-as-judge systems involve subjective tasks. It remains unclear whether the findings on self-preference in verifiable domains transfer to subjective evaluation settings, which substantially limits the paper’s practical contribution.

3. The evaluation is somewhat incomprehensive. The authors compute SPR over the entire $\mathcal D$ and analyze LSPR and HSPP on the $\mathcal D_{diff}$. However, the subset where evaluatees are both correct or both wrong, say $\mathcal D_{agree}$, is not analyzed in detail. I think this subset is particularly important for understanding the model’s self-preference that comes out of recognizing their own production when correctness provides no differentiating signals. Examining behavior on $\mathcal D_{agree}$ would offer a more complete view of self-preference behavior beyond what can be inffered from $\mathcal D_{diff}$ alone.

4. My biggest concern is that the claim in Line 269, “the degree of self-preference in stronger models is primarily driven by the objective quality of their outputs (evident by their higher judge accuracy)”, is an overclaim. What the paper actually demonstrates is a correlation, but this relationship may be affected by multiple confounding factors. The authors aim to conduct a systematic cross-family and cross-scale evaluation, which is a good point. But this design inevitably confound the relationship between generation accuracy and evaluation accuracy, making it unclear whether the upward trend in SPR arises from higher generation quality (a legitimate bias), or from identity/stylistic effects, where a model may recognize its own outputs. The latter one can increase SPR either legitimately or harmfully depending on the competitor’s quality. Alsom the authors state that stronger models tend to exhibit both greater harmful self-preference (Line 359) and mostly legitimate self-preference (Line 269). What I can derive from these phenomenon is that stronger models simply show greater self-preference overall, and their higher legitimacy ratio is possibily a byproduct of their higher task accuracy. Since they are correct often, their self-preference *appears* justified, which weakens the causal interpretation of these findings.
A more convincing design would involve comparing models of similar parameter scales or conducting analysis after controlling for generation accuracy to disentangle these effects.

5. Although this work is the first to explicitly measure self-preference on verifiable tasks, filling a gap in the existing literature, its contributions remain largely observational. I also note that several recent studies have already begun to mitigate or correct harmful self-preference, such as “Breaking the Mirror: Activation-Based Mitigation of Self-Preference in LLM Evaluators” and “Limits to Scalable Evaluation at the Frontier: LLM-as-Judge Won’t Beat Twice the Data.” It would strengthen the paper to review and discuss these works in the related literature.

**Questions:**

Seek weaknesses.

---

> ### Author Response · Authors · 2025-11-24
> **Response to Reviewer PZ79**
>
> We sincerely appreciate the reviewer’s constructive feedback!
>
> *Q1*
>
> > … in Line 252-256, the authors interpret correlations between model accuracy and self-preference as causal evidence that stronger models “recognize correct answers” or “identify bugs,” yet no qualitative analyses or case studies are provided to support these assertions. … the paper should include some examples showing how the LLM-as-judge identifies or explains specific errors in outputs to substantiate this claim.
>
> *A1*
>
> To substantiate the claim, we present a case study on the MBPP+ code generation task in **Appendix F**. The results demonstrate that performant models that generate correct code can **accurately identify bugs** and errors in peer code and **provide legitimate preference judgments** that align with the objective quality of the responses.
>
>
> *Q2*
>
> > This results have limited generalizability … most real-world applications of LLM-as-judge systems involve subjective tasks. It remains unclear whether the findings on self-preference in verifiable domains transfer to subjective evaluation settings …
>
> *A2*
>
> We have **expanded our experiments on a large-scale, real-world subjective dataset from LMArena** and added a **new, dedicated Section 4** in our revised manuscript. The results in **Figure 5** demonstrate that the same correlations – between task accuracy, judge accuracy, self-preference, and legitimate self-preference – persist under subjective tasks, suggesting **our findings extend beyond verifiable domains**.
>
> *Q3*
>
> > The evaluation is somewhat incomprehensive … the subset where evaluatees are both correct or both wrong, say D_agree, is not analyzed in detail. I think this subset is particularly important for understanding the model’s self-preference that comes out of recognizing their own production when correctness provides no differentiating signals …
>
> *A3*
>
> We agree that the $D_{\text{agree}}$ region would be informative for understanding stylistic/recognition-driven self-preference when correctness offers no signal. Following the suggestion, in **Section C.1**, **we conduct experiment decoupling self-preference evaluation into correctness-driven ($D_{\text{diff}}$) and style-driven ($D_{\text{agree}}$) subsets**. The results in **Figure 8** show $D_{\text{diff}}$ exhibits a consistently larger self-preference ratio, and as model scales, the gap between  $D_{\text{diff}}$ and $D_{\text{agree}}$ widens substantially: **stronger models exhibit greater spikes in self-preference when answers disagree**, while their **preference based on style alone grows at a much slower rate.** This suggests their self-preference is **increasingly driven by the objective quality** of the response **rather than stylistic bias**.
>
> Coupled with the observation that legitimate self-preference ratio correlates with task performance (Figure 4), these results indicate that **when responses are objectively different, stronger models prefer themselves more (Figure 8)**, and **when stronger models prefer themselves, it is more likely that their responses are indeed objectively better (Figure 4)**.
>
> *Q4*
>
> > … The authors aim to conduct a systematic cross-family and cross-scale evaluation, which is a good point. But this design inevitably confound the relationship between generation accuracy and evaluation accuracy … A more convincing design would involve comparing models of similar parameter scales or conducting analysis after controlling for generation accuracy to disentangle these effects.
>
> *A4*
>
> We added a new **capability-controlled analysis** in **Section C.2** to compare evaluator–evaluatee pairs with similar parameter scales and task accuracies. The results (**Tables 16–18**) of the controlled comparisons **produce the same trends** as the shared-evaluatee setting: stronger evaluators consistently record higher LSPR. The results reinforce our findings by **disentangling potential confounders arising from evaluator–evaluatee capability mismatches**.
>
> *Q5*
>
> > … several recent studies have already begun to mitigate or correct harmful self-preference, such as “Breaking the Mirror: Activation-Based Mitigation of Self-Preference in LLM Evaluators” and “Limits to Scalable Evaluation at the Frontier: LLM-as-Judge Won’t Beat Twice the Data.” It would strengthen the paper to review and discuss these works in the related literature.
>
> *A5*
>
> Thanks for bringing these works to our attention. We have incorporated a discussion of both studies into **Section 6** of our revised manuscript.
>
> Please let us know if you have any further questions – we would be happy to address them. If our responses adequately address your concerns, we would greatly appreciate your consideration in updating the score. Thank you again for your time and thoughtful review!

---

> > ### Comment · Reviewer_PZ79 · 2025-11-28
> >
> > I thank the authors for providing the additional explanations, which effectively addressed most of my concerns, and I would be very happy to increase my score. Unfortunately, due to recent technical issues on OpenReview and their consequences, the system no longer allows me to modify my score. I sincerely regret this situation, but I would be glad to update my score accordingly if an opportunity becomes available later.

---

### Official Review · Reviewer_k99Z · 2025-10-26

**Soundness:** 3
**Presentation:** 3
**Contribution:** 2
**Rating:** 4
**Confidence:** 3

**Summary:**

This paper studies self-preference (LLMs favoring their own outputs when acting as evaluators) on verifiable benchmarks so the authors can distinguish legitimate self-preference (preferring objectively better self-outputs) from harmful self-preference (preferring objectively worse self-outputs). They run a controlled, large-scale pairwise evaluation across many model families and sizes (Llama, Qwen, Gemma, Mistral, Phi, GPT, DeepSeek distilled models) on three verifiable tasks: MATH500 (math reasoning), MMLU (factual knowledge), and MBPP+ (code generation). They define metrics—SPR (self-preference ratio), LSPR (legitimate self-preference ratio), and HSPP (harmful self-preference propensity)—and show: (1) stronger models prefer themselves more, but much of that preference is legitimate (Figures 2–4); (2) when stronger models are wrong as generators they more often still prefer their own incorrect outputs (HSPP increases with capability; Figure 5); (3) generating reasoning (CoT / long CoT) before making a verdict substantially reduces harmful self-preference (Figure 6). The paper gives practical recommendations for safer LLM-as-judge use (model pre-assessment, inference-time scaling, routing to specialized evaluators).

**Strengths:**

* Clear, useful framing. Distinguishing legitimate vs harmful self-preference is a simple idea but crucial for interpreting prior findings; the paper operationalizes this clearly (SPR / LSPR / HSPP).
* Controlled experiments. Same evaluatee set across judges, multiple model families and scales, and three distinct verifiable domains (math, factual, code) improves generality.
* Concrete empirical findings. Strong models are better evaluators and often prefer their own (legitimate), but they also exhibit higher harmful self-preference when they err, a non-obvious safety concern emphasized with numbers and per-model results.

**Weaknesses:**

* Further interpretation and insight for self-preference. The paper would benefit from a deeper exploration of why self-preference arises and under what conditions it becomes harmful versus legitimate. For instance, is harmful self-preference linked to model confidence calibration, output style similarity, or internal alignment mechanisms? Providing diagnostic analyses (e.g., correlation with log-probabilities or similarity metrics) could make the findings more interpretable and actionable.

* Self-preference in subjective tasks. While using verifiable benchmarks is a strength, which allows clean separation between legitimate and harmful self-preference, it remains important to validate whether the observed patterns hold in subjective or open-ended tasks, where evaluation bias is more subtle and consequential. Such tasks are inherently more prone to hidden self-preference because ground-truth correctness is ambiguous. The authors could test robustness by leveraging annotated pairwise samples from LMArena or similar datasets containing subjective human judgments. This would help assess whether the same mitigation strategies (e.g., CoT reasoning) generalize to evaluators on creative or conversational domains. I would be open to raising my overall score if such experiments or discussions were added.

**Questions:**

See the weakness section.

---

> ### Author Response · Authors · 2025-11-24
> **Response to Reviewer k99Z**
>
> We thank the reviewer for the constructive suggestion!
>
> *Q1*
>
> > … The paper would benefit from a deeper exploration of why self-preference arises and under what conditions it becomes harmful versus legitimate. … Providing diagnostic analyses (e.g., correlation with log-probabilities or similarity metrics) could make the findings more interpretable and actionable.
>
> *A1*
>
> We appreciate the suggestion, and in **Section C.4**, we added such diagnostic analyses by **examining judgment confidence** (the evaluator’s assigned probability on its verdict to prefer its own response). We show that, in **Table 20 (also included below)**, **instances of harmful self-preference consistently exhibit a lower confidence than general self-preferred cases**. This pattern indicates that when the evaluator mistakenly favors its own incorrect output, it does so with weaker internal certainty. Notably, this confidence gap between SPR and HSPP correlates strongly with model capability: **as models scale, they become better calibrated**. This **latent uncertainty signal in larger models** offers a promising avenue for automatically detecting and mitigating harmful self-preference.
>
> | **Model**         | **Set** | **Avg. probability of 𝒥\* = j₁ (Prefer itself)** |
> |-------------------|---------|--------------------------------------------------|
> | Llama-3.2-3B  | SPR     | 31.14                                            |
> |                   | HSPP    | 30.60                                            |
> | Llama-3.1-8B  | SPR     | 57.01                                            |
> |                   | HSPP    | 48.81                                            |
> | Llama-3.1-70B | SPR     | 83.83                                            |
> |                   | HSPP    | 64.99                                            |
>
> *Q2*
>
> > … While using verifiable benchmarks is a strength, which allows clean separation between legitimate and harmful self-preference, it remains important to validate whether the observed patterns hold in subjective or open-ended tasks … The authors could test robustness by leveraging annotated pairwise samples from LMArena or similar datasets containing subjective human judgments …
>
> *A2*
>
> We have **expanded our experiments on the suggested large-scale, real-world subjective dataset from LMArena** and added a **new, dedicated Section 4** in our revised manuscript. The results as shown in **Figure 5** demonstrate the same correlations – between task accuracy, judge accuracy, self-preference, and legitimate self-preference – persist under subjective tasks, suggesting **our findings extend beyond verifiable domains**.
>
>
> Please let us know if you have any further questions – we would be happy to address them. If our responses adequately address your concerns, we would greatly appreciate your consideration in updating the score. Thank you again for your time and effort!

---

> > ### Comment · Reviewer_k99Z · 2025-11-24
> >
> > Thanks for authors' response. The added experiment and analysis address my concern and I have updated my score accordingly.

---

### Official Review · Reviewer_6Z6C · 2025-10-31

**Soundness:** 3
**Presentation:** 3
**Contribution:** 2
**Rating:** 4
**Confidence:** 4

**Summary:**

- Investigates whether LLM judges' self-preference reflects genuine quality or harmful bias using verifiable math, factual QA, and code tasks, formalizing position-robust aggregation J^* and a self-preference ratio.
- Finds that stronger models tend to prefer their own outputs largely in line with objective correctness, while harmful self-preference is captured by HSPP.
- Shows that longer CoT before verdicts mitigates HSPP and offers practical guidance for LLM-as-judge pipelines.

**Strengths:**

- Comprehensive coverage across evaluator models and benchmarks with careful handling of position bias and ties.
- Clear visualizations and definitions that show relationships between capability and self-preference.
- Actionable mitigation signal: longer Chain-of-Thought reduces HSPP with a known token-budget tradeoff.
- Evidence aligns across model families and scales, supporting robust empirical patterns.

**Weaknesses:**

- Limited external validity to subjective or multi-criteria evaluations.
- Possible coupling when a judge evaluates its own generations, suggesting a withheld-generator check.
- Narrow decoding settings; robustness to sampling choices is not fully explored.
- Limited theoretical analysis and few mitigation avenues beyond longer reasoning, plus missing statistical significance reporting.

**Questions:**

- How do results extend to subjective or multi-attribute domains and to formats like scalar scoring or ranking?
- How do sampling-based decoding choices affect SPR and HSPP, and does longer CoT still help under higher response diversity?
- Can harmful self-preference be isolated by fixing the judge and evaluating on responses from withheld generators?
- What linguistic or family-level style correlates predict harmful self-preference, and what mechanisms drive it?

---

> ### Author Response · Authors · 2025-11-24
> **Response to Reviewer 6Z6C**
>
> We thank the reviewer for the thoughtful comments!
>
> *Q1*
>
> > Limited external validity to subjective or multi-criteria evaluations.
> > How do results extend to subjective or multi-attribute domains and to formats like scalar scoring or ranking?
>
> *A1*
>
> We have **expanded our experiments on a large-scale, real-world subjective dataset from LMArena** and added **a new, dedicated Section 4** in our revised manuscript.
> The results in **Figure 5** demonstrate that the same correlations – between task accuracy, judge accuracy, self-preference, and legitimate self-preference – persist under subjective tasks, suggesting **our findings extend beyond verifiable domains**.
>
> *Q2*
>
> > Possible coupling when a judge evaluates its own generations, suggesting a withheld-generator check.
>
> *A2*
>
> We would like to clarify that **this setup is precisely the problem setting our work aims to study**: self-preference bias in LLM evaluators. This evaluation protocol is **the standard and widely adopted practice** in automatic evaluation benchmarks, including AlpacaEval 2 [1], where a judge model routinely compares an evaluatee’s response against the judge’s own output; and also, in self-improvement literature, the models continuously judge their own outputs [2][3].
> Our goal is precisely to analyze this widely used and practically important evaluation setting. Moreover, our experiments span a broad range of model families and scales, including both open-source and proprietary models, to ensure the generality and robustness of our findings.
>
> *Q3*
>
> > Narrow decoding settings; robustness to sampling choices is not fully explored … missing statistical significance reporting ... How do sampling-based decoding choices affect SPR and HSPP, and does longer CoT still help under higher response diversity?
>
> *A3*
>
> To evaluate the robustness of long CoT mitigation for harmful self-preference propensity under
> different decoding configurations, we **experiment with different decoding temperatures {0.4, 0.6, 0.8}** and report results in **Table 19 (also included below)**. As shown, long CoT reasoning consistently achieves lower HSPP scores (with statistical significance; † indicates *p* < 0.01) across all tested temperatures, **demonstrating the stability of its mitigating effect**. More details are provided in **Section C.3** in our revised manuscript.
>
> | **Method**                     | **HSPP**        |
> |-------------------------------|------------------|
> | No reasoning                  | 29.53            |
> | Long CoT reasoning        |                  |
> | &emsp;– *Temperature = 0.4*   | 11.87†           |
> | &emsp;– *Temperature = 0.6*   | 15.34†           |
> | &emsp;– *Temperature = 0.8*   | 15.51†           |
>
> *Q4*
>
> > Limited theoretical analysis and few mitigation avenues beyond longer reasoning … What linguistic or family-level style correlates predict harmful self-preference, and what mechanisms drive it?
>
> *A4*
>
> We added an analysis in **Section C.4** by examining judgment confidence (the evaluator’s assigned probability on its verdict to prefer its own response), and show in **Table 20 (also included below)** that **instances of harmful self-preference consistently exhibit a lower confidence than general self-preferred cases**. This pattern indicates that when the evaluator mistakenly favors its own incorrect output, it does so with weaker internal certainty. Notably, this confidence gap between SPR and HSPP correlates strongly with model capability: **as models scale, they become better calibrated**. This **latent uncertainty signal in larger models** offers a promising avenue for automatically detecting and mitigating harmful self-preference.
>
>
> | **Model**         | **Set** | **Avg. probability of 𝒥\* = j₁ (Prefer itself)** |
> |-------------------|---------|--------------------------------------------------|
> | Llama-3.2-3B  | SPR     | 31.14                                            |
> |                   | HSPP    | 30.60                                            |
> | Llama-3.1-8B  | SPR     | 57.01                                            |
> |                   | HSPP    | 48.81                                            |
> | Llama-3.1-70B | SPR     | 83.83                                            |
> |                   | HSPP    | 64.99                                            |
>
> We are more than happy to address any additional questions you might have. If our responses adequately address your concerns, we would greatly appreciate your consideration in updating the score. Thanks again for your time!
>
>
> --
>
> [1] Dubois et al. "*Length-controlled alpacaeval: A simple way to debias automatic evaluators.*" COLM 2024.
>
> [2] Madaan et al. "*Self-refine: Iterative refinement with self-feedback.*" NeurIPS 2023.
>
> [3] Yuan et al. "*Self-rewarding language models.*" ICML2024.

---

### Meta-Review · Area_Chair_pqHe · 2026-01-02

**Summary:**

This paper studies self-preference bias in LLM-based evaluators and asks whether such preference reflects harmful bias or simply higher output quality from stronger models. The key idea is to move beyond subjective tasks by evaluating self-preference on verifiable benchmarks (math reasoning, factual QA, and code generation), allowing the authors to distinguish between "legitimate" self-preference (preferring a correct self-generated answer) and "harmful" self-preference (preferring an incorrect one). The paper presents large-scale experiments across model families and sizes, and further explores mitigation via inference-time reasoning. Reviewers generally agreed that the experimental setup is careful and that the analysis is thorough. However, there was disagreement about the overall level of novelty and the extent to which the results change existing understanding of LLM-as-a-judge behavior.

**Reviewer Concerns:**

The rebuttal addressed several clarification and completeness issues:
- The authors clarified the distinction between legitimate and harmful self-preference and how these quantities are measured.
- Additional analyses and discussion were provided to connect the results to subjective evaluation settings.
- The mitigation results using inference-time reasoning were better contextualized.

However, the core concerns raised by multiple reviewers remain:
- The primary contribution is empirical and diagnostic rather than methodological. While the decomposition is useful, it refines existing observations about self-preference rather than introducing a fundamentally new framework or capability.
- The mitigation strategy (e.g. longer CoT or reasoning traces) is intuitive and incremental, and does not amount to a new technique.
- As a result, the paper's main conclusions sharpen existing understanding but do not substantially advance the state of the art.

Overall, while the rebuttal improved clarity and presentation, it did not change the reviewers' underlying assessment that the contribution is limited in scope.

**Reviewer Scores:**

- For 6Z6C, while the rebuttal clarified several points and strengthened the presentation, it did not introduce new contributions that would materially change this reviewer's assessment. This reviewer would likely maintain a similar 4 score, maybe a 5.
- k99Z explicitly stated that the rebuttal addressed their main concerns and that they updated their score accordingly.
- PZ79 raised the strongest concerns in the initial review, particularly around novelty and the extent to which the results go beyond prior work. In their response to the rebuttal, this reviewer stated that most of their concerns had been addressed and that they would have increased their score if the system had allowed it. They didn't specify what they'd increase their score to, so lacking that assessment I can only assume that they're go from a 2 to a 4 at best.

This paper was truly borderline in my eyes.

---

### Decision · Program_Chairs · 2026-01-26

Reject